# A Pre-Trained Model Customization Framework for Accelerated PET/MR Segmentation of Abdominal Fat in Obstructive Sleep Apnea

**DOI:** 10.3390/diagnostics15243243

**Published:** 2025-12-18

**Authors:** Valentin Fauveau, Heli Patel, Jennifer Prevot, Bolong Xu, Oren Cohen, Samira Khan, Philip M. Robson, Zahi A. Fayad, Christoph Lippert, Hayit Greenspan, Neomi Shah, Vaishnavi Kundel

**Affiliations:** 1Biomedical Engineering and Imaging Institute (BMEII), Icahn School of Medicine at Mount Sinai, New York, NY 10029, USA; 2Digital Health & Machine Learning, Hasso Plattner Institute, University of Potsdam, 14469 Potsdam, Germany; 3Division of Pulmonary, Critical Care, and Sleep Medicine, Department of Medicine, Icahn School of Medicine at Mount Sinai, New York, NY 10029, USA; 4Hasso Plattner Institute for Digital Health at Mount Sinai, Icahn School of Medicine at Mount Sinai, New York, NY 10029, USA; 5School of Biomedical Engineering, Faculty of Engineering, Tel-Aviv University, Tel-Aviv 39040, Israel

**Keywords:** obstructive sleep apnea (OSA), visceral adipose tissue (VAT), subcutaneous adipose tissue (SAT), cardiovascular disease (CVD), artificial intelligence (AI), deep learning (DL), transfer learning, RadImageNet (RIN), positron emission tomography (PET), magnetic resonance imaging (MRI), biomarker

## Abstract

**Background**: Accurate quantification of visceral (VAT) and subcutaneous adipose tissue (SAT) is critical for understanding the cardiometabolic consequences of obstructive sleep apnea (OSA) and other chronic diseases. This study validates a customization framework using pre-trained networks for the development of automated VAT/SAT segmentation models using hybrid positron emission tomography (PET)/magnetic resonance imaging (MRI) data from OSA patients. While the widespread adoption of deep learning models continues to accelerate the automation of repetitive tasks, establishing a customization framework is essential for developing models tailored to specific research questions. **Methods**: A UNet-ResNet50 model, pre-trained on RadImageNet, was iteratively trained on 59, 157, and 328 annotated scans within a closed-loop system on the Discovery Viewer platform. Model performance was evaluated against manual expert annotations in 10 independent test cases (with 80–100 MR slices per scan) using Dice similarity coefficients, segmentation time, intraclass correlation coefficients (ICC) for volumetric and metabolic agreement (VAT/SAT volume and standardized uptake values [SUVmean]), and Bland–Altman analysis to evaluate the bias. **Results**: The proposed deep learning pipeline substantially improved segmentation efficiency. Average annotation time per scan was 121.8 min (manual segmentation), 31.8 min (AI-assisted segmentation), and only 1.2 min (fully automated AI segmentation). Segmentation performance, assessed on 10 independent scans, demonstrated high Dice similarity coefficients for masks (0.98 for VAT and SAT), though lower for contours/boundary delineation (0.43 and 0.54). Agreement between AI-derived and manual volumetric and metabolic VAT/SAT measures was excellent, with all ICCs exceeding 0.98 for the best model and with minimal bias. **Conclusions**: This scalable and accurate pipeline enables efficient abdominal fat quantification using hybrid PET/MRI for simultaneous volumetric and metabolic fat analysis. Our framework streamlines research workflows and supports clinical studies in obesity, OSA, and cardiometabolic diseases through multi-modal imaging integration and AI-based segmentation. This facilitates the quantification of depot-specific adipose metrics that may strongly influence clinical outcomes.

## 1. Introduction

Obesity is a national and global epidemic, with forecasts suggesting that almost two in three adults will be overweight or obese by the year 2050 [1,2]. Both overweight and obesity result in a high burden of adverse health-related cardiometabolic conditions [3], particularly obstructive sleep apnea (OSA), where 40–60% of OSA cases are attributable to excess weight [4,5]. OSA affects an estimated 1 billion people worldwide [6], where recurrent upper airway obstruction leads to sleep fragmentation, daytime sleepiness, and an elevated cardiovascular disease risk [7].

Central abdominal obesity is among the strongest risk factors for OSA and other cardiometabolic diseases [8,9]. Central obesity can be further classified into visceral adipose tissue (VAT) and subcutaneous adipose tissue (SAT). This distinction is of clinical relevance, as excess VAT is considered metabolically unhealthy and is independently associated with increased risk for metabolic syndrome, diabetes mellitus [10,11,12,13,14], atherosclerosis, and cardiovascular disease (CVD) [15,16,17,18]. Traditional metrics of obesity, such as body mass index (BMI) and waist circumference, fail to account for differences in body fat composition and abdominal fat. Therefore, an accurate quantification of VAT and SAT is essential for evaluating cardiometabolic risk in OSA and other chronic disease states. Studies have investigated a variety of software tools for both automated and semi-automated segmentation of abdominal adipose tissue using computed tomography (CT) or magnetic resonance imaging (MRI) [19,20,21,22]. To our knowledge, few studies have validated automated segmentation for assessments of depot-specific abdominal fat volume and metabolic activity using hybrid ^18^F-fluorodeoxyglucose (FDG) positron emission tomography (PET)/MRI [23]. FDG is a commonly used radiotracer and glucose analog that accumulates in inflammatory foci. Studies have demonstrated increased FDG uptake in VAT versus SAT (indicating proinflammatory macrophage activity), suggesting increased adipose tissue inflammation [24,25].

Our pilot study [26] established the feasibility of using hybrid FDG PET/MRI to quantify visceral and subcutaneous fat volume and metabolic activity in patients with OSA using manual segmentations. To expedite the labor-intensive process of manual annotation, this study aimed to develop and validate a deep-learning-based model for automated multi-slice segmentation of SAT and VAT in patients with OSA, using hybrid FDG PET/MRI and leveraging existing datasets. The primary objective was to reduce segmentation time relative to manual annotation while rigorously evaluating model performance against expert references for both volumetric and metabolic adipose metrics.

## 2. Materials and Methods

### 2.1. Patient Population

Our study utilized images from two prospective sleep cohorts to train our deep learning models. Patients were recruited from the Mount Sinai Health System in New York City.

Cohort 1 (NIH [R01HL143221], Mount Sinai IRB 18-00557) recruited 200 treatment-naïve OSA patients for the assessment of imaging biomarkers of atherosclerosis and subclinical CVD using PET/MRI (obtained from the base of the skull to the upper thighs) between 2018 and 2022. Recruited patients met the following inclusion criteria: (1) newly diagnosed OSA (respiratory disturbance index [RDI] of >5 events/hour sleep on in-laboratory or home sleep apnea testing); (2) age > 21 and <80 years; (3) no previous OSA treatment; and (4) the presence of one or more cardiovascular risk factors, including hypertension, hyperlipidemia, diabetes, smoking, and obesity. Patients with central sleep apnea, sleep hypoventilation, history of CVD events, insulin-dependent diabetes, a cancer diagnosis within the last year, and individuals who were pregnant or breastfeeding were excluded. Of the 200 recruited patients, 180 patients completed baseline imaging with PET/MRI.

Cohort 2 (NIH K23HL161324, Mount Sinai IRB 22-00838, 2022-present) recruited adults from the Sleep and Internal Medicine clinics with mild OSA and presence of one or more cardiovascular risk factors (as above) to evaluate the association between objective sleep metrics and imaging biomarkers of atherosclerosis and visceral obesity using PET/MRI. Exclusion criteria were similar to those outlined for cohort 1. Hybrid PET/MRI with ^18^F-FDG radiotracer was used to evaluate abdominal obesity metrics (SAT and VAT volume and metabolic activity).

### 2.2. Image Acquisition Protocol

Both cohorts included simultaneous (hybrid) PET and MRI acquisitions with ^18^F-FDG radiotracer to quantify abdominal adipose volume and metabolic activity. High-resolution MRI provided structural information, and PET provided functional assessment. Image acquisition and analysis were conducted at the Mount Sinai BioMedical Engineering and Imaging Institute (BMEII).

MRI acquisition parameters included a slice thickness of 2.09 mm, in-plane resolution of 291 × 360 pixels, and voxel dimensions of 1.39 × 1.39 × 2.09 mm. Images were acquired using a 3D axial composed MR-based attenuation-corrected (MRAC), controlled aliasing in parallel imaging (CAIPI) fat-phase sequence without contrast, with participants performing a breath hold during acquisition.

Co-registration of MRI and PET images was used for quantitative assessments of adipose tissue volume and metabolic activity, as previously published [26]. All acquired PET list-mode data were histogrammed into a single data frame and later reconstructed with an Ordinary Poisson—Ordered Subsets Expectation Maximization (OP-OSEM) algorithm (3 iterations, 21 subsets) to derive the respective ^18^F-FDG 3D PET images.

### 2.3. Manual Annotation Task for Deep Learning Model Training

Expert manual annotations were performed on the MRI scans using the OsiriX platform (Pixmeo Sarl, Bernex, Switzerland; www.osirix-viewer.com, accessed on 12 December 2025). The methodology for manual segmentation of abdominal adipose tissue contours is detailed in our previously published work and has been validated by peer researchers [27,28,29]. In that pilot study, we demonstrated excellent inter-reader reproducibility and reliability of our measurement techniques for volumetric and metabolic adipose tissue values [26].

Each patient scan consisted of 80–100 MR slices, where 90.65 ± 10.81 slices were annotated on average (range 61–121 slices). The internal and external SAT contours for each slice were manually segmented, along with segmentations of highly metabolic organs such as the kidneys and spinal bone marrow. Manual segmentations were performed from the pelvic floor (L4–L5 interspace) to the diaphragm. The resulting ROIs included the external SAT abdominal contour (EXT; external SAT), the internal SAT abdominal contour (INT; internal SAT), and segmentations of the kidneys and spinal bone marrow, which were marked for exclusion (EXC; kidneys and spinal bone marrow) (Figure 1).

### 2.4. Volumetric and Metabolic Quantification of Adipose Tissue

The total SAT volume was computed using the INT and EXT abdominal masks. Binary masks were generated to delineate the SAT compartment on each slice, which were then summed and multiplied by the voxel size to determine total SAT volume per scan (Figure 2).

The total VAT volume was more challenging due to its complex anatomical boundaries. The visceral fat is distributed around internal organs within the abdominal cavity, requiring precise anatomical knowledge and careful attention during segmentations, which makes the process time-consuming. To streamline this process, we defined the VAT region as the volume within the boundaries of INT minus EXC. The resulting mask was then refined by applying a slice-level mask threshold, excluding all low-intensity pixels to retain only high-intensity regions corresponding to adipose tissue (Figure 2). For each slice, the threshold value was computed based on the mean and standard deviation of MRI pixel intensities. This threshold was applied to generate a binary mask, delineating the visceral fat region. Finally, the total VAT volume was determined by summing the VAT masks across all slices and multiplying by the voxel size.

To evaluate metabolic activity, PET images were co-registered with structural MRI to facilitate the extraction of SUVmean from predefined regions of interest (SAT and VAT volumes). The co-registration process was simplified by the PET/MR imaging protocol, which enables simultaneous acquisition of inherently aligned images. Volumetric segmentations were subsequently used to extract and evaluate the accuracy of SUVmean values. SAT and VAT PET activity were quantified by resampling the respective ROI coordinates to PET resolution using nearest neighbor interpolation. Finally, slice-wise SUVmean values were computed and averaged across all slices to obtain global SAT and VAT SUVmean estimates.

### 2.5. Discovery Viewer Platform

For this study, we leveraged the Discovery Viewer (DV) platform (version 0.3), which serves as an advanced interface for medical image analysis, designed to streamline the integration of artificial intelligence (AI) models into research workflows. This platform facilitates real-time interaction with AI-generated predictions, allowing users to visualize, annotate, and refine these predictions efficiently [30].

At the core of this research, integrated within the DV framework, is the UNet-ResNet50 model pre-trained on RadImageNet (RIN), a deep learning architecture specifically optimized for medical image segmentation. This model combines two powerful components: the UNet architecture [31], which is well known for its ability to delineate anatomical structures with high accuracy, and ResNet50 [32], a convolutional neural network pre-trained on RadImageNet, a large-scale medical imaging dataset comprising over 1.35 million MRI, CT, ultrasound (US), and X-ray (XR) images spanning a wide range of anatomical regions [33].

By combining UNet with a ResNet50 that has already “learned” from such a vast and diverse medical dataset, the model can start off with strong foundational knowledge. This helps provide a strong initialization, enabling more efficient learning and improved performance on downstream segmentation tasks, particularly in scenarios with limited labeled training data. The integration of UNet’s segmentation capabilities with the feature extraction power of RadImageNet-trained ResNet50 facilitates accelerated convergence and enhanced segmentation accuracy in medical imaging applications.

### 2.6. Model Development and Continuous Annotation Workflow

We adopted an iterative training strategy to progressively improve model performance while minimizing annotation burden. Model development followed an annotation–correction workflow. The RIN UNet-ResNet50 architecture was first trained on 59 manually annotated PET/MRI scans (V1). Expert reviewers corrected the resulting segmentations, which were incorporated into a larger training set (157 scans) to develop V2. After additional corrections, V3 was trained on 328 scans. This staged approach allowed expert feedback at each step to refine segmentation quality while substantially reducing manual annotation time (Figure 3). The summary of the training parameters can be observed in Table 1.

### 2.7. Model Performance Metrics

Segmentation accuracy and efficiency were evaluated first. An independent expert reviewer manually annotated 10 patient scans in OsiriX. These manual segmentations were compared with predictions from each model iteration (V1, V2, V3) for the same 10 patient scans on DV. Evaluation metrics included processing time and segmentation accuracy (for mask and contour/boundary delineation), with the latter quantified using the Dice similarity coefficient (DSC).

Second, volumetric and metabolic reproducibility were examined on the same validation cohort. VAT and SAT volumetric and metabolic values were obtained from both manual and automated methods for each model iteration (V1, V2, V3). Agreement was quantified using Bland–Altman plots and intraclass correlation coefficients (ICCs), thereby assessing the reproducibility of the automated measurements.

### 2.8. Statistical Analysis: Efficiency, Segmentation Accuracy, and Validation of Volumetric and Metabolic Measures

Times were recorded between the start and the end of the segmentation process. Time efficiency was measured by calculating the average time per scan on the validation cohort (*n* = 10), comparing manual versus AI-assisted segmentations (V3) and fully automated AI segmentations. We present the processing times in boxplot format, compared using the Wilcoxon signed-rank test.

To evaluate the accuracy of the AI-assisted segmentation, we employed dice scores [34], a widely used metric for quantifying the overlap between AI-generated segmentations and expert manual annotations. The Dice Score ranges from 0 to 1, where values closer to 1 indicate a higher degree of agreement, highlighting the reliability of the AI-driven approach. To ensure a thorough assessment, the DSC was calculated not only for the entire segmentation mask but also specifically for the contours, which were obtained by performing a morphological gradient (dilation—erosion) using a 3 × 3 kernel. These contours are critical for evaluating the fine-grained accuracy of the border’s segmentation (Figure 4).

ICC and Bland–Altman plots were used to evaluate agreement and bias between VAT and SAT volumetric and metabolic values derived from AI-assisted and expert manual segmentations.

## 3. Results

### 3.1. Time Analysis

The proposed method dramatically improved segmentation efficiency by significantly reducing annotation time. The AI-assisted method required, on average, 31.8 min, showing a substantial reduction compared to manual annotation, which required 121.8 min. The AI model alone completed processing in an average of 1.2 min per scan (Figure 5).

### 3.2. Segmentation Performance

Segmentation accuracy of the AI models for VAT and SAT regions on MRI was evaluated using DSC for INT, EXT, and EXC masks and contours (Table 2). The three model iterations (V1, V2, V3) were compared against manual annotations (*n* = 10). For mask areas, the mean DSC across iterations was 0.98 for INT, 0.98 for EXT, and 0.84 for EXC regions. For contours, mean DSC values were lower, averaging 0.43 for INT, 0.54 for EXT, and 0.35 for EXC regions.

### 3.3. Validation of Volumetric and Metabolic Measures

Agreement between AI-derived and manually annotated volumetric and metabolic VAT/SAT measurements was assessed using ICC (*n* = 10). The analysis demonstrated excellent reliability, with ICC values consistently high (≥0.94 for all models and ≥0.98 for manual segmentation vs. V3), indicating robust agreement between AI and manual methods (Table 3). The Bland–Altman plots (Figure 6, Table 4) revealed no significant systematic bias for the average SUVmean (V3 VAT SUVmean −0.013 ± 0.016, 95% limits of agreement [LOA] −0.04 to 0.018; V3 SAT SUVmean 0.003 ± 0.003, 95% LOA −0.002 to 0.009) or the V3 VAT/SAT volume ratio (−0.012 ± 0.05, 95% LOA −0.1 to 0.08). A small bias was observed for volumetric measurements (V3 VAT volume 117 ± 75 cm^3^, 95% LOA −31 to 266 cm^3^; V3 SAT volume 93 ± 135 cm^3^, 95% LOA −171 to 359 cm^3^), reflecting a slight overestimation attributable to model contour imprecisions; however, this difference of approximately 100 cm^3^ is considered clinically negligible relative to the total adipose tissue volumes. For context, the mean SAT volume is 5513 cm^3^, and the mean VAT volume is 3138 cm^3^. The mean VAT bias of 117 cm^3^ represents only 3.7% of the mean VAT volume, and the mean SAT bias of 93 cm^3^ is only 1.7% of the mean SAT volume.

Notably, this bias progressively decreased across model iterations, with V3 demonstrating the highest agreement with manual measurements (Table 4).

## 4. Discussion

In this study, we developed and validated a custom segmentation AI model for visceral and subcutaneous adipose tissue through a continuous transfer-learning framework in the Discovery Viewer platform. Our model substantially reduced segmentation time and improved efficiency compared to conventional manual annotation methods. Evaluation using Dice similarity coefficients demonstrated that the model effectively identified regions of interest compared to expert manual segmentations, even from its initial iteration (V1), highlighting the strength of our transfer-learning approach in accelerating the manual annotation process. Moreover, the SAT/VAT volumes and SAT/VAT SUVmean values derived from our model’s segmentations showed high agreement and overall limited bias compared with those obtained via manual annotation, supporting the quantitative reliability of our approach. Nonetheless, opportunities remain for further refinement to achieve more precise boundary delineation.

Although the contour-level DSC values were relatively low, the results are acceptable given the high mask DSC scores and excellent volumetric ICCs. The results demonstrate that the model reliably identifies the overall fat regions necessary for accurate volume and SUVmean estimation, which are the primary outcomes of this study. The strong volumetric agreement indicates that contour discrepancies are localized and do not substantially affect total volume calculations. Nonetheless, the suboptimal boundary precision suggests room for improvement. While the UNet–ResNet50 backbone pre-trained on RadImageNet provides a strong initialization, incorporating edge-aware or boundary-refinement loss functions (e.g., Boundary IoU, Hausdorff distance, or distance-transform loss) may enhance fine contour delineation. We highlight these strategies as promising directions for future development of the platform, alongside continued dataset expansion and exploration of post-processing methods to further improve segmentation precision and clinical utility.

Automated and semi-automated methods for adipose tissue segmentation have been applied across CT, MRI, and a few PET/MRI platforms [20]. Linder et al. validated the semiautomatic tool segfatMR against manual multi-slice MRI annotations of VAT and SAT in patients with obesity, showing strong correlation and a two-fold improvement in processing speed [21]. Küstner et al. applied large-scale deep learning efforts for adipose segmentation using whole-body MRI in a cohort of ~1000 participants, showing good Dice scores [35]. More recently, one study automated whole-body PET/MRI for quantification of volume and metabolic activity metrics across different tissue classes, including abdominal fat, achieving moderate–high Dice scores for VAT and SAT [36]. Together, these studies highlight progress toward automated volumetric fat quantification but also underscore challenges in annotation burden and reproducibility, with a promising role for deep learning-based segmentation.

While AI-driven advancements provide effective solutions for automating segmentation tasks, many existing open-source AI models are either inaccessible to clinicians and researchers or lack the flexibility required for clinical customization. As such, there is a critical need for intuitive, adaptable AI pipelines that support broader implementation in both clinical and research environments. For example, publicly available segmentation models did not meet the requirements for our specific cohort study, which aimed to segment all MR slices across the abdominal area: from the pelvic floor (L4–L5 interspace) to the diaphragm. Additionally, in our study, our segmentations excluded breast adipose tissue to prevent overestimation of abdominal subcutaneous fat. We also excluded highly metabolic tissues such as the kidneys and spinal bone marrow, which could otherwise falsely elevate the VAT FDG signal. Therefore, we argue that rather than pursuing the development of a universally generalizable model, improving accessibility to foundation models that can be customized for specific research needs represents a more practical solution to accelerate research tasks. Each research project has unique requirements; therefore, a fully generalizable model is unlikely to satisfy all cases. Instead, enabling flexible frameworks that allow researchers to easily adapt models to their particular applications can lead to meaningful improvements in research workflow efficiency.

Our study presents a closed-loop AI training pipeline designed for the continuous refinement of deep learning models for medical imaging, in which model-generated predictions are systematically reviewed and incorporated back into the training dataset to improve performance accuracy with each iteration. We highlight the effectiveness of transfer learning to achieve strong segmentation performance, addressing the challenge of limited labeled datasets. Our study also highlights the importance of customizability over generalizability and explores the use of a transfer-learning framework for the customization of deep learning models to expedite real research needs.

This approach presents a more practical pathway for applying AI models to real-world research, where study-specific requirements often limit the utility of broadly generalizable models. In our case, this customization facilitated the segmentations of adipose tissue for the analysis of the fat metabolic activity of OSA patients using PET/MRI data. This capability has direct implications for the study of conditions such as obesity, OSA, and cardiometabolic diseases, where depot-specific adipose burden may strongly influence cardiovascular risk and treatment response. Specifically in the context of sleep apnea research, VAT/SAT quantification holds critical clinical implications, given the strong association of OSA with elevated cardiovascular disease risk. Visceral obesity is a well-established risk factor for cardiovascular disease and may act as a critical mediator in the pathophysiological link between OSA and cardiovascular outcomes [24]. Therefore, by improving the efficiency and accessibility of VAT/SAT assessments, the presented framework allows an easy and accelerated development of AI models that have the potential to enhance the understanding of the complex interplay between OSA and body composition and assess fat depot-specific treatment responses to therapy.

## 5. Limitations

Although our proposed method provided efficient results, there are a couple of limitations to acknowledge. Manual revisions were still required at each iteration of the AI model development to ensure optimal accuracy. Despite this, our approach accelerated the annotation process by a factor of four compared to manual segmentation alone and is expected to continue improving as additional cases are annotated and the models are further refined. The slight underperformance observed in certain metrics, such as the Dice score for excluded ROIs or the contours, implies that this may potentially affect volumetric accuracy downstream, indicating the opportunity for further optimization. Importantly, however, these discrepancies did not meaningfully affect the volumetric or metabolic outcome measures for VAT and SAT, which were the key clinical variables for cardiometabolic risk stratification.

We also acknowledge that a closed-loop training framework using expert-corrected data has the potential to introduce sample-selection bias or overfitting toward the annotator’s segmentation style. Although annotator-specific bias cannot be fully excluded, results from our prior pilot work assessing inter-reader reliability between two independent readers demonstrated high human–human agreement for VAT and SAT metrics, with all ICC values for VAT/SAT volume and SUVmean exceeding 0.9, indicating excellent inter-reader reliability [31]. This level of consistency suggests that any annotator-driven bias introduced during model refinement is likely limited to the task evaluated. Additionally, this bias can further be mitigated in future studies through larger training datasets engaging multiple annotators.

The model’s generalizability across diverse datasets and imaging conditions remains to be validated. Nonetheless, our primary objective was not to achieve generalizability but rather to maximize annotation efficiency, enabling rapid model development with minimal manual effort. This strategy allows researchers, including those without AI expertise, to accelerate research workflows and collectively contribute to AI model refinement.

## 6. Conclusions

Our study demonstrates the effectiveness of our continuous AI model training approach in significantly reducing annotation time while maintaining high segmentation accuracy for VAT/SAT quantification using the PET/MRI platform. Leveraging the Discovery Viewer platform, we integrated AI model development directly into the annotation workflow, enabling rapid validation, refinement, and continuous performance improvement through a closed-loop training system. The success of this approach highlights the importance of accessible, collaborative AI tools that allow researchers and healthcare professionals to contribute to model development and validation.

## Figures and Tables

**Figure 1 diagnostics-15-03243-f001:**
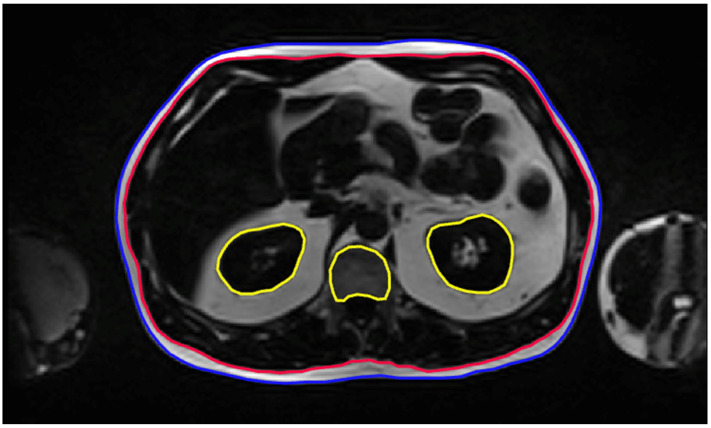
Annotation Task. Representative MRI slices with manual annotations used to train the AI segmentation model. External (EXT, blue) and internal (INT, red) abdominal contours delineate the subcutaneous adipose tissue on each slice, while exclusion regions (EXC, yellow) outline the kidneys and spinal bone marrow.

**Figure 2 diagnostics-15-03243-f002:**
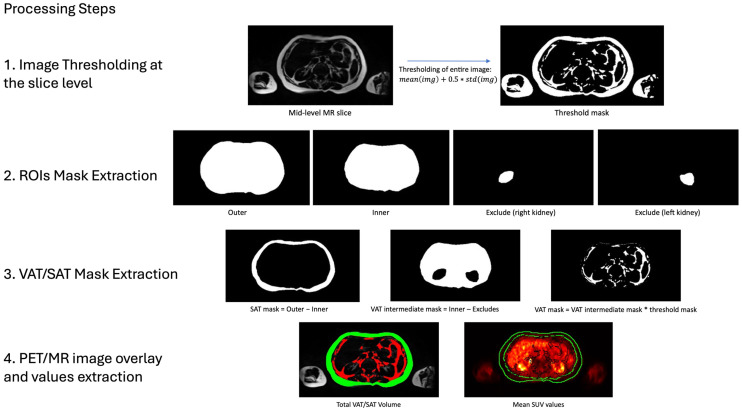
Extraction of VAT/SAT Volumetric and Metabolic Values. This figure illustrates the four-step process used to derive volumetric and metabolic measurements of VAT and SAT. First, MRI images were thresholded at the slice level to isolate high-intensity pixel regions. Second, extracted segmentations were used to define SAT and VAT areas: SAT was calculated by subtracting the internal abdominal ROI (INT) from the external abdominal ROI (EXT), while VAT was determined by subtracting the exclusion ROIs (EXC) from the internal abdominal area (INT). Third, to isolate fat-specific regions within VAT and SAT, the calculated threshold mask was applied to the segmented areas. Fourth, total VAT (red) and SAT (green) volumes were obtained by summing the corresponding fat masks across all slices. For metabolic analysis, VAT and SAT mean standardized uptake values (SUVs) were derived by co-registering the PET images to MRI and averaging the voxel values within each segmented volume.

**Figure 3 diagnostics-15-03243-f003:**
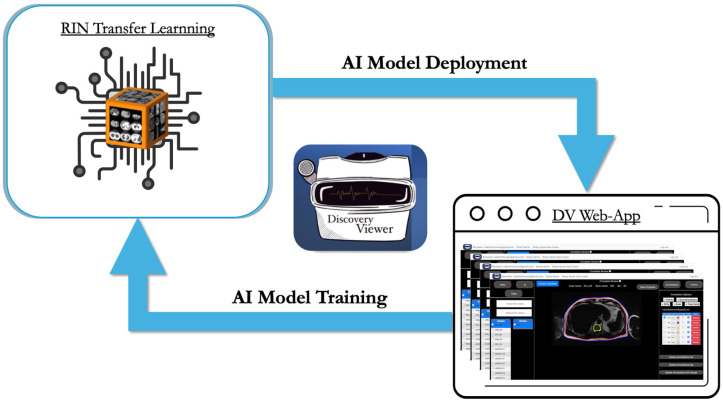
Continuous closed-loop AI model training strategy. This continuous closed-loop framework enables users to link their annotations directly to various pre-trained networks, allowing for the ongoing refinement of customized AI models. Trained models are subsequently deployed within the platform to accelerate labor-intensive and time-consuming tasks. In this study, 59 manually annotated cases were used to develop the initial model (V1) based on a pre-trained RIN U-Net with a ResNet-50 backbone. Model V1 was then refined using 157 annotated cases to create V2, followed by further refinement with 328 cases to develop the final V3 model.

**Figure 4 diagnostics-15-03243-f004:**
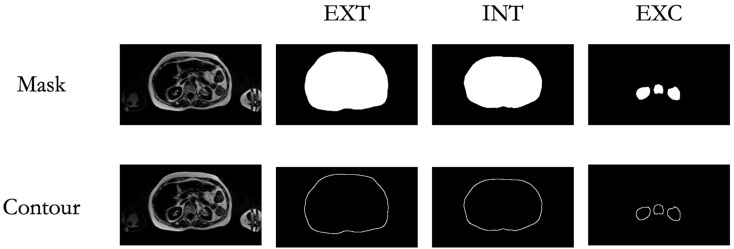
Mask and Contour Extraction. Visualization of the extracted masks and contours for the targeted regions of interest (ROIs), including the external abdominal contour (EXT), internal abdominal contour (INT), and exclusion areas (EXC).

**Figure 5 diagnostics-15-03243-f005:**
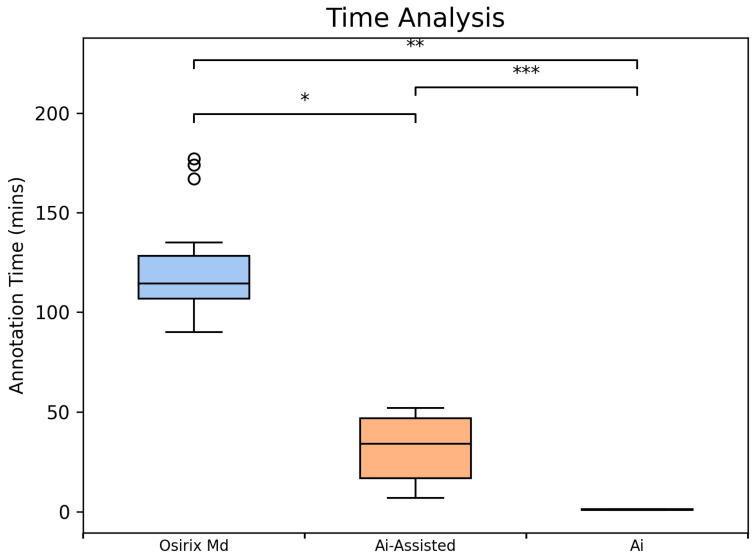
Comparative Analysis of Segmentation Time using Box Plots (*n* = 10 scans). Average segmentation time per scan, including manual segmentation time in Osirix (blue, 96.8 min per scan), AI-assisted (V3; orange, 31.8 min per scan) and automated AI (green, 1.14 min per scan) segmentation times in the Discovery Viewer (DV) platform. Significance indicators: *p* < 0.05 (*), *p* < 0.01 (**), *p* < 0.001 (***).

**Figure 6 diagnostics-15-03243-f006:**
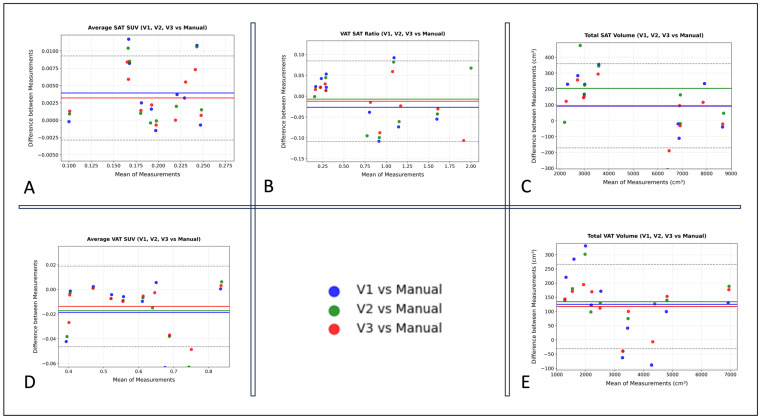
Bland–Altman plots comparing VAT and SAT volumetric and metabolic values derived from manual versus automated annotations across the three AI model versions: V1, V2, and V3 (*n =* 10). (**A**) Average SAT SUV (SUVmean). (**B**) VAT/SAT Volume Ratio. (**C**) Total SAT Volume (cm^3^). (**D**) Average VAT SUV (SUVmean). (**E**) Total VAT Volume (cm^3^).

**Table 1 diagnostics-15-03243-t001:** Core training parameters. The table provides an overview of the principal training parameters used across the three model versions (V1, V2, and V3) for segmenting each ROI: the external SAT contour (EXT), internal SAT contour (INT), and exclusion mask for highly metabolic organs (EXC; kidneys and spinal bone marrow).

Parameters\Model Version	V1	V2	V3
**Imaging Specification**	High-Resolution MRI	High-Resolution MRI	High-Resolution MRI
**Architecture**	UNet-ResNet50	UNet-ResNet50	UNet-ResNet50
**Weight Initialization**	RadImageNet	V1	V2
**Number of Training Scans**	59	157	328
**Input Dimension**	224 × 224	224 × 224	224 × 224
**Normalization Method**	Z-Score	Z-Score	Z-Score
**Loss Function**	Binary Cross entropy	Binary Cross entropy	Binary Cross entropy

**Table 2 diagnostics-15-03243-t002:** Assessment of Segmentation Performance Using Dice Scores (mask and contours). This table presents Dice Similarity Coefficient (DSC) values comparing manual annotations to model versions V1, V2, and V3 (*n* = 10). The mask DSC—critical for downstream analyses—remains consistently high across all comparisons, reflecting substantial volumetric overlap. Contour DSC, which evaluates boundary accuracy, shows limited improvement with each successive model iteration.

Manual Annotations vs.	V1		V2		V3	
	Mask	Contour	Mask	Contour	Mask	Contour
**INT**	0.98	0.41	0.98	0.43	0.98	0.45
**EXT**	0.98	0.55	0.98	0.53	0.99	0.55
**EXC**	0.82	0.35	0.85	0.34	0.83	0.37

**Table 3 diagnostics-15-03243-t003:** Agreement of volumetric and metabolic measurements between automated and manual annotations. This table reports the volumetric and metabolic measurements for VAT and SAT derived from MRI-based segmentations and PET-based metabolic activity. Values represent the mean of 10 assessed samples. Intraclass correlation coefficients (ICCs) between each model and the manual segmentations demonstrate consistently high agreement, indicating strong concordance across segmentation methods. The variability in adipose tissue distributions among OSA patients is reflected by the large standard deviations observed for VAT and SAT volumes. These key metrics, combined with their metabolic activity, are essential for a robust understanding of the association between fat distribution, sleep apnea severity and cardiovascular disease.

Outcome	Manual vs. AI Model Outcome Values				Intraclass Correlation Coefficient (ICC)		
Mean (SD)	V1	V2	V3	Manual	V1 vs. Manual	V2 vs. Manual	V3 vs. Manual
**Total SAT Volume (cm^3^)**	5540 (2805)	5610 (2864)	5574 (2822)	5513 (2971)	0.99	0.99	0.99
**Total VAT Volume (cm^3^)**	3281 (1456)	3295 (1520)	3290 (1520)	3138 (1572)	0.99	0.99	0.99
**Average SAT SUVmean**	0.24 (0.08)	0.24 (0.08)	0.24 (0.08)	0.242 (0.08)	0.96	0.94	0.98
**Average VAT SUVmean**	0.71 (0.26)	0.71 (0.25)	0.71 (0.25)	0.746 (0.26)	0.99	0.99	0.99
**VAT/SAT Ratio**	0.74 (0.46)	0.75 (0.5)	0.75 (0.48)	0.728 (0.49)	0.97	0.98	0.99

**Table 4 diagnostics-15-03243-t004:** Bland–Altman Summary statistics comparing Automated vs. Manual Segmentations Across Model Iterations. The table presents the mean and standard deviation of the bias (Prediction—Manual) and the corresponding 95% limits of agreement for each model iteration.

Variable	AI Model	Mean Bias ± SD	95% Limits of Agreement (Lower, Upper)
**VAT SUVmean**	V1	−0.018 ± 0.026	(−0.07, 0.03)
	V2	−0.017 ± 0.02	(−0.05, 0.02)
	V3	−0.013 ± 0.016	(−0.04, 0.018)
**SAT SUVmean**	V1	0.004 ± 0.004	(−0.004, 0.01)
	V2	0.013 ± 0.02	(−0.04, 0.06)
	V3	0.003 ± 0.003	(−0.002, 0.009)
**VAT Volume (cm^3^)**	V1	125 ± 129	(−128, 379)
	V2	133 ± 82	(−27, 295)
	V3	117 ± 75	(−31, 266)
**SAT Volume (cm^3^)**	V1	91 ± 223	(−346, 528)
	V2	203 ± 323	(−429, 836)
	V3	93 ± 135	(−171, 359)
**VAT/SAT Volume Ratio**	V1	−0.026 ± 0.088	(−0.2, 0.146)
	V2	−0.007 ± 0.06	(−0.12, 0.11)
	V3	−0.012 ± 0.05	(−0.1, 0.08)

## Data Availability

The data used to train the models is not readily available because it includes information from an ongoing study in which some participants did not consent to sharing their individual data with external parties in the informed consent. Requests to access the datasets may be directed to Dr. Kundel and Mr. Fauveau. The annotation and modeling platform is presently available as an internal proof-of-concept at the Icahn School of Medicine at Mount Sinai. A public, cloud-hosted release is planned to support broader collaboration, including community-driven updates and peer review of predictive medical models. For early access, contact Valentin Fauveau (valentin.fauveau@mssm.edu).

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
