# Peer review of "A Pre-Trained Model Customization Framework for Accelerated PET/MR Segmentation of Abdominal Fat in Obstructive Sleep Apnea"

_diagnostics, 2025, doi:10.3390/diagnostics15243243_

Round 1

Reviewer 1 Report

Comments and Suggestions for Authors

This paper presents a pre-trained model customization framework for accelerated segmentation of abdominal fat using PET/MR imaging in obstructive sleep apnea (OSA) patients. The authors employ a RadImageNet pre-trained UNet–ResNet50 architecture integrated into the Discovery Viewer (DV) platform for iterative refinement, achieving remarkable time reduction from 122 minutes (manual) to 1.2 minutes (AI) per scan with high Dice coefficients (≈0.98) and excellent ICC (>0.98) for both VAT and SAT quantification. The pipeline enables efficient integration of volumetric and metabolic adipose tissue assessment within hybrid PET/MRI, which is of great value for cardiometabolic and sleep medicine research.

The study is methodologically robust, and the proposed workflow is well-motivated, integrating transfer learning effectively for clinical adaptation.

  1. The segmentation framework is technically sound and well-implemented, but there are a few advanced considerations that could further strengthen its methodological rigor. While the UNet–ResNet50 backbone pre-trained on RadImageNet provides a strong initialization, the contour-level Dice coefficients suggest that boundary precision remains suboptimal. This could be addressed by integrating edge-aware or boundary-refinement losses (e.g., Boundary IoU, Hausdorff distance, or distance-transform loss) during training, which have been shown to improve fine delineations in medical segmentation tasks.
  2. While method comprehensive, the methods section is quite dense. Simplifying or tabulating parameters (e.g., imaging specifications, training dataset sizes, and model settings) would improve readability.
  3. Dice scores for contours remain low (0.35–0.55). Discuss more concretely how boundary refinement could be improved (e.g., edge-aware loss, post-processing filters).
  4. Since the DV tool is currently internal, it would help to describe when and how the public release will be available or how researchers can request early access.

Author Response

Date: 12/01/2025

Manuscript ID: diagnostics-3998045

Title: A Pre-Trained Model Customization Framework for Accelerated PET/MR Segmentation of Abdominal Fat in Obstructive Sleep Apnea

The authors would like to thank the reviewer for the constructive comments and suggestions. Each comment has been carefully considered, and appropriate revisions have been made. The original comments from the reviewer are shown in black, and the detailed responses are shown in blue.

Reviewer #1:

Comment 1: The segmentation framework is technically sound and well-implemented, but there are a few advanced considerations that could further strengthen its methodological rigor. While the UNet–ResNet50 backbone pre-trained on RadImageNet provides a strong initialization, the contour-level Dice coefficients suggest that boundary precision remains suboptimal. This could be addressed by integrating edge-aware or boundary-refinement losses (e.g., Boundary IoU, Hausdorff distance, or distance-transform loss) during training, which have been shown to improve fine delineations in medical segmentation tasks.

Response: We appreciate this insightful recommendation. While it is not feasible to integrate these loss functions into our existing results, we have included this consideration in the Discussion Section Paragraph 2 as a meaningful direction for future platform improvements. Enabling users to employ edge-aware or boundary-refinement loss functions that could help mitigate the boundary-precision challenges observed in our study.

Comment 2: While method comprehensive, the methods section is quite dense. Simplifying or tabulating parameters (e.g., imaging specifications, training dataset sizes, and model settings) would improve readability.

Response: Thank you for your suggestion. To improve readability, we have reduced the density of the Methods Section and added a comprehensive table listing the imaging specifications, dataset sizes, and key model training parameters.

Comment 3: Dice scores for contours remain low (0.35–0.55). Discuss more concretely how boundary refinement could be improved (e.g., edge-aware loss, post-processing filters).

Response: As stated in our response to Comment 1, the incorporation of edge-aware or boundary-refinement loss functions presents a promising strategy for improving contour delineation, and this has been addressed in the Discussion Section (paragraph 2) as future improvements of the platform. While our study evaluated the impact of dataset expansion on model performance, future research could further investigate the integration of boundary-focused loss functions and post-processing techniques to enhance efficiency and precision.

Comment 4: Since the DV tool is currently internal, it would help to describe when and how the public release will be available or how researchers can request early access.

Response: At present, the DV tool is internal, and still under development and testing - not yet ready for public release. A pilot deployment across all the Mount Sinai network is planned for January 2026. We have added a software availability statement for anyone who might be interested in an early access by contacting the corresponding authors.

Reviewer 2 Report

Comments and Suggestions for Authors

Training in a loop with the expert-corrected data may bring about sample selection bias or overfitting against the annotators’ segmentation style.

What merits more conversation is how generalizable the model is to different institutions, scanners, or annotation schemas.

For the whole mask, this results in high DSC (0.98), but low contour DSC scores (0.43 and 0.543 ).

It implies that the model has difficulty obtaining precise boundaries, which may affect volumetric accuracy downstream in more complex tasks.

However, the test set is too small (only 10 scans) while training data set large (59 → 157 → 328 scans), which may also affect the generalization.

A more extensive validation or external test sample would confirm the results.

Author Response

Date: 12/01/2025

Manuscript ID: diagnostics-3998045

Title: A Pre-Trained Model Customization Framework for Accelerated PET/MR Segmentation of Abdominal Fat in Obstructive Sleep Apnea

The authors would like to thank the reviewer for the constructive comments and suggestions. Each comment has been carefully considered, and appropriate revisions have been made. The original comments from the reviewer are shown in black, and the detailed responses are shown in blue.

Reviewer #2: *We have consolidated related comments and provide a unified response below.*

Comment: Training in a loop with the expert-corrected data may bring about sample selection bias or overfitting against the annotators’ segmentation style.

Response: Thank you for the valuable comment. We have incorporated the corresponding revisions into the manuscript.

We acknowledge that expert-corrected training may introduce annotator-specific bias. To provide context, in the Methods Section2.3 we referenced our previous publication which demonstrates excellent inter-reader reproducibility and reliability of our measurement techniques for volumetric and metabolic adipose tissue values [32]. We further elaborate on this in the Limitations Section 5 paragraph 2, adding that bias can further be mitigated through larger datasets and multiple annotators.

Comment: For the whole mask, this results in high DSC (0.98), but low contour DSC scores (0.43 and 0.543 ). It implies that the model has difficulty obtaining precise boundaries, which may affect volumetric accuracy downstream in more complex tasks.

Response: Thank you for this comment. As brought up by other Reviewers, we have addressed this point by adding the corresponding clarification to the Discussion Section, paragraph 2, and Limitations Section, paragraph 1.

Comment: However, the test set is too small (only 10 scans) while training data set large (59 → 157 → 328 scans), which may also affect the generalization. A more extensive validation or external test sample would confirm the results. What merits more conversation is how generalizable the model is to different institutions, scanners, or annotation schemas.

Response: We thank the reviewer for this comment. Regarding the small test sample size (10 scans), we agree that a larger or external validation cohort would provide a clearer assessment of generalizability. However, our main point was to emphasize on the importance of customizability, the ability to rapidly tailor models to specific research needs, rather than generalizability. Ultimately, we sought to demonstrate the utility of foundation models and a user-friendly fine-tuning platform for enabling a wide range of clinical research applications. We  elaborate on this point as well in the last paragraph of the Limitations Section 5, paragraph 3.

Reviewer 3 Report

Comments and Suggestions for Authors

The following points should be addressed to improve clarity and strengthen the paper's quantitative claims:

A. Clarification of Segmentation Accuracy Discrepancy

The Results section reports a very high Dice similarity coefficient (DSC) for the segmentation masks (0.98 for both VAT and SAT), but a substantially lower DSC for the contours/boundary delineation (0.43 and 0.54)7. This discrepancy is expected in segmentation tasks, but its implications need to be more clearly reconciled with the "excellent" agreement reported for volumetric and metabolic metrics (ICC $\ge$ 0.98)8.

  • Requested Change: In the Discussion (Section 4), explicitly explain that the poor contour DSC is acceptable because the high mask DSC and excellent volumetric ICCs demonstrate that the AI model accurately identifies the overall region of fat to ensure accurate volume and mean metabolic activity (SUVmean) calculation, which is the primary research outcome. The high volumetric agreement suggests the contour errors are localized and do not significantly impact the total volume calculation.

B. Quantifying the Volumetric Bias in Context

The Bland-Altman analysis for V3 reveals a slight bias (overestimation) in volumetric measurements (VAT: $117 \pm 75 \text{ cm}^{3}$; SAT: $93 \pm 135 \text{ cm}^{3}$) which the authors state is "considered clinically negligible relative to the total adipose tissue volumes"9.

  • Requested Change: To fully support this claim, please quantify the relative magnitude of the bias in the Results (Section 3.3) or Discussion. For context, the mean SAT Volume is $5513 \text{ cm}^{3}$ and the mean VAT Volume is $3138 \text{ cm}^{3}$10. The mean VAT bias ($117 \text{ cm}^{3}$) is only $\sim 3.7\%$ of the mean VAT volume, and the mean SAT bias ($93 \text{ cm}^{3}$) is $\sim 1.7\%$ of the mean SAT volume. Stating these percentages will make the argument for "clinically negligible" much more rigorous.

3. Specific/Minor Edits

  1.  

    Abstract (Lines 44-45)11: The final sentence of the abstract is slightly verbose. Consider splitting it for better flow and impact:

    • Suggestion: "Our framework streamlines research workflows and supports clinical studies in obesity, OSA, and cardiometabolic diseases through multi-modal imaging integration and AI-based segmentation. This facilitates the quantification of depot-specific adipose metrics that may strongly influence clinical outcomes."

  2.  

    Introduction (Line 60)12: The acronyms VAT and SAT are introduced here but are not officially parenthesized (VAT) and (SAT). They are defined in the Keywords section 13and Abstract14. For consistency, ensure the first usage in the main body of the text clearly defines the acronyms.

    • Suggestion: "...visceral adipose tissue (VAT) and subcutaneous adipose tissue (SAT)."

  3.  

    Methods - Annotation Task (Section 2.3)15: The ROIs are defined as EXT, INT, and EXC. While defined in the figure caption16, consider a small, parenthetical reminder in the text for the reader's convenience (e.g., "...exclusion (EXC, kidneys and spinal bone marrow) (Figure 1).").

Author Response

Date: 12/01/2025

Manuscript ID: diagnostics-3998045

Title: A Pre-Trained Model Customization Framework for Accelerated PET/MR Segmentation of Abdominal Fat in Obstructive Sleep Apnea

The authors would like to thank the reviewer for the constructive comments and suggestions. Each comment has been carefully considered, and appropriate revisions have been made. The original comments from the reviewer are shown in black, and the detailed responses are shown in blue.

Reviewer #3:

Comment 1: Clarification of Segmentation Accuracy Discrepancy

The Results section reports a very high Dice similarity coefficient (DSC) for the segmentation masks (0.98 for both VAT and SAT), but a substantially lower DSC for the contours/boundary delineation (0.43 and 0.54)7. This discrepancy is expected in segmentation tasks, but its implications need to be more clearly reconciled with the "excellent" agreement reported for volumetric and metabolic metrics (ICC  0.98).

Requested Change: In the Discussion (Section 4), explicitly explain that the poor contour DSC is acceptable because the high mask DSC and excellent volumetric ICCs demonstrate that the AI model accurately identifies the overall region of fat to ensure accurate volume and mean metabolic activity (SUVmean) calculation, which is the primary research outcome. The high volumetric agreement suggests the contour errors are localized and do not significantly impact the total volume calculation.

Response: Thank you for this helpful suggestion. We have addressed it by adding the corresponding clarification to the Discussion Section, paragraph 2.

Comment 2: Quantifying the Volumetric Bias in Context

The Bland-Altman analysis for V3 reveals a slight bias (overestimation) in volumetric measurements (VAT: 117 +/- 75 cm3 and SAT: 93 +/- 135 cm3) which the authors state is "considered clinically negligible relative to the total adipose tissue volumes".

Requested Change: To fully support this claim, please quantify the relative magnitude of the bias in the Results (Section 3.3) or Discussion. For context, the mean SAT Volume is 5513 cm3 and the mean VAT Volume is 3138 cm3. The mean VAT bias (117 cm3) is only 3.7% of the mean VAT volume, and the mean SAT bias (93 cm3) is 1.7% of the mean SAT volume. Stating these percentages will make the argument for "clinically negligible" much more rigorous.

Response: We appreciate this excellent suggestion and fully agree that including these percentages would offer a clearer and more quantitative representation of the volumetric biases. We added the modification in the Results, Section 3.3.

Comment 3: Specific/Minor Edits

1. Abstract (Lines 44-45)11: The final sentence of the abstract is slightly verbose. Consider splitting it for better flow and impact:

Suggestion: "Our framework streamlines research workflows and supports clinical studies in obesity, OSA, and cardiometabolic diseases through multi-modal imaging integration and AI-based segmentation. This facilitates the quantification of depot-specific adipose metrics that may strongly influence clinical outcomes."

Response: Thank you for your suggestion, we agreed with it, and we have made the change in the Abstract.

2. Introduction (Line 60): The acronyms VAT and SAT are introduced here but are not officially parenthesized (VAT) and (SAT). They are defined in the Keywords section and Abstract. For consistency, ensure the first usage in the main body of the text clearly defines the acronyms.

Suggestion: "...visceral adipose tissue (VAT) and subcutaneous adipose tissue (SAT)."

Response: Thank you for your suggestion. We agree with it and have implemented the changes in the Introduction.

3. Methods - Annotation Task (Section 2.3). The ROIs are defined as EXT, INT, and EXC. While defined in the figure caption, consider a small, parenthetical reminder in the text for the reader's convenience (e.g., "...exclusion (EXC, kidneys and spinal bone marrow) (Figure 1).").

Response: Thank you for the suggestion. We have added reminders of the acronym definitions within the figures to improve clarity in Methods Section 2.3.
